# Clinical Impact of Dual Time Point ^18^F-Fluorodeoxyglucose Positron Emission Tomography/Computed Tomography Fusion Imaging in Pancreatic Cancer

**DOI:** 10.3390/cancers14153688

**Published:** 2022-07-28

**Authors:** Takahiro Einama, Yoji Yamagishi, Yasuhiro Takihata, Fukumi Konno, Kazuki Kobayashi, Naoto Yonamine, Ibuki Fujinuma, Takazumi Tsunenari, Keita Kouzu, Akiko Nakazawa, Toshimitsu Iwasaki, Eiji Shinto, Jiro Ishida, Hideki Ueno, Yoji Kishi

**Affiliations:** 1Department of Surgery, National Defense Medical College, Saitama 359-8513, Japan; einama0722@ndmc.ac.jp (T.E.); yy0uji1982@gmail.com (Y.Y.); y_takihata@ndmc.ac.jp (Y.T.); nippon.mamoru.f@gmail.com (F.K.); harinezumi08@gmail.com (K.K.); desertstylecarl@yahoo.co.jp (N.Y.); fjnmibk811@gmail.com (I.F.); barca3444@gmail.com (T.T.); dj27qd.t01312kk@gmail.com (K.K.); akiko_happy_0207@yahoo.co.jp (A.N.); toiwasaki@ndmc.ac.jp (T.I.); shinto@ndmc.ac.jp (E.S.); ueno_surg1@ndmc.ac.jp (H.U.); 2Tokorozawa PET Diagnostic Imaging Clinic, Saitama 359-1124, Japan; ishida@toko-pet.or.jp

**Keywords:** pancreatic cancer, dual time point PET/CT, SUVmax1, SUVmax2, ΔSUVmax%

## Abstract

**Simple Summary:**

In pancreatic cancer, recurrence rates after surgery remain high. The ability to identify patients at risk of early recurrence before surgery will contribute to the selection of treatment strategies. We examined the value of preoperative dual time point (DTP) ^18^F-fluorodeoxyglucose positron emission tomography/computed tomography fusion imaging (FDG PET/CT) as a predictor of early recurrence in or the outcomes of patients with pancreatic cancer. The results showed that DTP FDG PET/CT may effectively predict relapse in patients, and the combination of SUVmax1 and ΔSUVmax% identified early recurrent patient groups more precisely than SUVmax1 alone.

**Abstract:**

We examined the value of preoperative dual time point (DTP) ^18^F-fluorodeoxyglucose positron emission tomography/computed tomography fusion imaging (FDG PET/CT) as a predictor of early recurrence or the outcomes in patients with pancreatic cancer. Standardized uptake values (SUVs) in DTP FDG PET/CT were performed as preoperative staging. SUVmax1 and SUVmax2 were obtained in 60 min and 120 min, respectively. ΔSUVmax% was defined as (SUVmax2 − SUVmax1)/SUVmax1 × 100. The optimal cut-off values for SUVmax parameters were selected based on tumor relapse within 1 year of surgery. Optimal cut-off values for SUVmax1 and ΔSUVmax% were 7.18 and 24.25, respectively. The combination of SUVmax1 and ΔSUVmax% showed higher specificity and sensitivity, and higher positive and negative predictive values for tumor relapse within 1 year than SUVmax1 alone. Relapse-free survival (RFS) was significantly worse in the subgroups of high SUVmax1 and high ΔSUVmax% (median 7.0 months) than in the other subgroups (*p* < 0.0001). The multivariate Cox analysis of RFS identified high SUVmax1 and high ΔSUVmax% as independent prognostic factors (*p* = 0.0060). DTP FDG PET/CT may effectively predict relapse in patients with pancreatic cancer. The combination of SUVmax1 and ΔSUVmax% identified early recurrent patient groups more precisely than SUVmax1 alone.

## 1. Introduction

Pancreatic cancer has a dismal prognosis, which is highlighted by the close relationship between disease incidence and mortality within 1 year. Surgery with curative intent is recommended for 15–20% of patients who present with resectable tumors. Fewer than 1 in 5 patients have early-stage disease amenable to potentially curative resection, and only 20% of these patients survive for 5 years [1,2,3]. The ability to identify patients at risk of early recurrence before surgery will contribute to the selection of optimal treatment strategies. In recent years, ^18^F-fluorodeoxyglucose positron emission tomography/computed tomography (FDG PET/CT) has been increasingly used to diagnose biological properties and stage and detect disease recurrence. FDG PET/CT enables the metabolic rate of glucose to be visualized in vivo [4]. FDG PET/CT is different from CT, which reflects anatomical structures, and MRI, which mainly reflects anatomical structures and diffusion. PET provides images of molecular and biological functions in vivo [5,6]. Glucose metabolism is generally enhanced in malignant tumors, and, thus, ^18^F-FDG uptake is increased. A high level of ^18^F-FDG accumulation on PET/CT is considered to represent the active form of tumor cells. Therefore, the maximum standardized uptake value (SUVmax) of primary cancer on FDG PET/CT may be used to estimate the outcomes of patients [7,8].

The sensitivity of FDG PET/CT for the detection of malignant lesions is high; however, FDG also accumulates in inflammatory lesions [9,10,11]. To overcome this limitation, previous studies demonstrated the efficacy of measuring ^18^F-FDG uptake levels at dual time points (DTP) [7,12,13]. The ^18^F-FDG uptake level at a later phase (2–3 h after the injection) is more likely to increase specifically in malignant tumors and decrease in benign tumors [14]. Correlations have been reported between ΔSUVmax% and malignant potential in lung cancer, lymphoma, and breast cancer [7,15,16,17]. However, PET/CT is costly and only available at a few institutions. Another drawback of FDG PET/CT is the false negative accumulation of SUV in patients with hyperglycemia. Hyperglycemia is associated with significantly reduced ^18^F-FDG up-take levels [18]. Diabetes mellitus is one of the risk factors for pancreatic cancer. Therefore, FDG-PET/CT has not been commonly applied as a tool for the preoperative evaluation of pancreatic cancer. The ability of ^18^F-FDG uptake measurements in DTP FDG PET/CT to predict the biological characteristics and outcomes of pancreatic cancer patients has not yet been examined. DTP evaluations of FDG uptake levels may overcome the issue of a decrease in the diagnostic accuracy of pancreatic cancer in patients with diabetes mellitus.

Therefore, we examined the predictive value of DTP FDG PET/CT for early recurrence in patients who underwent surgical resection for pancreatic cancer. We also investigated the efficacy of DTP evaluations of FDG uptake levels as a preoperative indicator of the outcomes.

## 2. Materials and Methods

The present study was performed with the approval of the Institutional Review Board of the National Defense Medical College, Tokorozawa, Japan (Approval No. 3038). All participants provided informed consent.

### 2.1. Patient Selection

Patients who underwent pancreatic resection for pancreatic cancer between January 2013 and April 2019 following preoperative DTP FDG PET/CT were selected. Pancreatic cancer was diagnosed based on cytological and/or pathological examinations before surgery. The comorbidity of diabetes mellitus was judged based on the medical history provided by each patient using a questionnaire survey on the day of admission or by blood examinations after admission to our hospital. Postoperative surveillance was performed through examinations of tumor markers every 3 months and CT every 6 months. PET/CT was also conducted to detect recurrence.

### 2.2. Quantification of ^18^F-FDG Uptake in Pancreatic Cancer

We performed FDG PET/CT at the Tokorozawa PET Diagnostic Imaging Clinic (Tokorozawa, Japan Biograph LSO Emotion, 3D model; Siemens, Munich, Germany). Patients fasted for at least 4 h before the examination. The first scan was performed 1 h after the intravenous administration of 3.7 Mbq/kg ^18^F-FDG. The first examination involved whole-body imaging from the head to the thigh for screening, while the second scan, which was conducted within 50–60 min of the first examination, focused on the abdomen for an evaluation of malignancy. After image reconstruction, 5 mm slice thickness, the region of interest (ROI) was placed in one area of the primary pancreatic cancer showing the highest uptake of ^18^F-FDG. SUV is defined as decay-corrected tissue activity divided by the injected dose per patient body and is calculated using the following formula: SUV = activity in ROI (MBq/mL)/injected dose (MBq = kg body weight)

SUVmax1 was obtained in the initial phase (60 min) and SUVmax2 in the delayed phase (120 min), and ΔSUVmax% was calculated using the following formula:ΔSUVmax% = [(SUVmax2 − SUVmax1)/SUVmax1] × 100

### 2.3. Histological Study

Tumor stages comprising the T, N, and M factors, the clinical stage, histological grade, and residual tumors were assigned according to the 8th Edition of the Union for International Cancer Control staging. A tumor diameter of 2 cm or less is designated as T1, of more than 2 cm, but no greater than 4 cm, as T2, of more than 4 cm at the greatest diameter as T3, and that involving the celiac axis, superior mesenteric artery, and/or common hepatic artery as T4. N0 refers to no lymph node metastases, N1 to metastases in 1 to 3 nodes, and N2 to metastases in 4 or more nodes. M0 refers to no distant metastases and M1 to existing distant metastases. We evaluated lymphatic permeation as positive or negative. 

### 2.4. Cut-Off Value to Predict Early Recurrence after Surgery

Receiver operating characteristic curves were drawn to select the optimal cut-off values for SUVmax1 and ΔSUVmax% that predict tumor relapse within 1 year of surgical resection. The Youden index [= sensitivity − (1 − specificity)] of each cut-off value was also calculated, and the value with the highest Youden index was selected as the optimal cut-off point. SUVmax1 and ΔSUVmax% values above and below the optimal cut-off were defined as high and low, respectively. The CA19-9 cut-off value was obtained using the same approach.

### 2.5. Statistical Analysis According to Clinicopathological Factors and Prognosis

The relationships between SUVmax parameters (SUVmax1, SUVmax2, and ΔSUVmax%) and clinicopathological factors were examined using the non-parametric Wilcoxon and Kruskal–Wallis tests. We used the Kaplan–Meier method to draw relapse-free survival (RFS) curves. Differences in survival curves were analyzed by the Log-rank test. A Cox proportional hazards model was used for univariate and multivariate analyses of RFS. The sensitivity, specificity, positive predictive value (PPV), negative predictive value (NPV), and accuracy of SUVmax1, ΔSUVmax%, and their combination for RFS were calculated. All differences were significant at *p* < 0.05. Statistical analyses were performed using JMP 14 (SAS Institute Inc., Cary, NC, USA).

## 3. Results

### 3.1. Patient Characteristics

During the study period, 146 patients underwent surgical resection for pancreatic cancer. Preoperative DTP FDG PET/CT was performed on 102 patients, 30 of whom were excluded from the analysis for the following reasons: (1) a history of preoperative chemotherapy (*n* = 19), (2) difficulty measuring SUVmax due to the insufficient accumulation of ^18^F-FDG (*n* = 8), and (3) other causes of death within 1 year of surgery (*n* = 3). The remaining 72 patients were examined. The median follow-up was 22.5 months (range 2.9–66.8 months).

The clinical and pathological profiles of patients are summarized in Table 1. Seventeen patients (24%) had a history of diabetes mellitus as a comorbidity. The medians and ranges of SUVmax1, mean SUVmax2, and ΔSUVmax% were 5.1 (1.7–22.1), 6.5 (1.9–25.6), and 24.6 (−13.9–84.4), respectively (Figure 1).

SUVmax1 was significantly lower in patients with diabetes mellitus than in those without diabetes mellitus, whereas ΔSUVmax% was similar between the two groups (Figure 2).

### 3.2. Setting of Optimal Cut-Off Values for Patient Prognostication and Accuracy of the Prediction of Relapse within 1 Year of Surgery

According to the Youden index, the optimal cut-off value for SUVmax1 was 7.18 with an area under the curve (AUC) of 0.59 (95% confidence interval (CI) 0.44–0.72) (Figure 3A). Patients were divided into the low SUVmax1 (<7.18) (*n* = 53) and high SUVmax1 groups (≥7.18) (*n* = 19). The optimal cut-off value for ΔSUVmax% was 24.25 with an AUC of 0.67 (95% CI 0.53–0.78) (Figure 3B). Patients were divided into the low ΔSUVmax% (<24.25) (*n* = 37) and high ΔSUVmax% groups (≥24.25) (*n* = 35). In addition, we divided patients using two approaches: (1) group A (*n* = 13), in which SUVmax1 and ΔSUVmax% were both high vs. group B (*n* = 59), and others, and (2) group C (*n* = 43), in which SUVmax1 and/or ΔSUVmax% were high vs. group D (*n* = 29), in which SUVmax1 and ΔSUVmax% were both low, because we hypothesized that the combination of SUVmax1 and/or ΔSUVmax% might more accurately predict the prognosis than the simple index such as SUVmax1 or ΔSUVmax%.

Among the 19 patients with SUVmax1 ≥ 7.18, recurrence was detected in 13 (68%) within 1 year of surgery, while 18 out of 53 (34%) patients with SUV < 7.18 exhibited early recurrence (*p* = 0.0091). Group A in comparison with group B showed higher specificity and PPV than high SUVmax1 alone. Group C in comparison with group D showed higher sensitivity and NPV than high SUVmax1 alone (Table 2).

### 3.3. Comparison of Clinical and Pathological Factors According to SUVmax1 and ΔSUVmax%

Clinicopathological parameters were compared using four different approaches according to high vs. low SUVmax1, high vs. low ΔSUVmax, SUVmax1, group A vs. group B, and group C vs. group D (Table 3). The incidence of lymph node metastases, lymphatic permeation, and the serum CA19-9 value significantly differed between the high and low SUVmax1 groups, whereas the distribution of the pathological T-factor, the R status, and the proportion of patients who completed adjuvant chemotherapy were similar. More patients completed adjuvant chemotherapy in the high ΔSUVmax% group than in the low ΔSUVmax% group (*p* = 0.0019). Group A was associated with a higher CA19-9 value (*p* = 0.0043) and group C with a lower frequency of completed adjuvant chemotherapy (*p* = 0.0076). In the serum CA19-9 value, group A vs. group B in comparison with the high vs. low SUVmax1 group showed higher sensitivity (61.5% vs. 47.4%) and NPV (90.4% vs. 80.8%). In the completed adjuvant chemotherapy, group C vs. group D in comparison with the low vs. high ΔSUVmax% group showed higher PPV (75.0% vs. 69.4%).

### 3.4. Comparison of Survival Curves

RFS significantly differed between the high and low SUVmax1 groups (*p* = 0.0004) (Figure 4A). A slight difference was observed in RFS between the high and low ΔSUVmax% groups (*p* = 0.058) (Figure 4B). RFS was significantly worse in group A than in group B (*p* < 0.0001) (Figure 4C) and in group C than in group D (*p* = 0.023) (Figure 4D). In diabetes mellitus patients, no significant difference was observed in RFS between the high and low ΔSUVmax% groups (*p* = 0.35) (Figure 4E), but the survival curve was similar to Figure 4B.

### 3.5. Univariate and Multivariate Analyses

Univariate analyses identified CA19-9 > 512 U/L, lymph node metastases, incomplete adjuvant chemotherapy, SUVmax1, and group A as independent predictive factors for worse RFS. Multivariate analyses including the former three clinicopathological parameters and either SUVmax1 or group A showed that SUVmax1 and group A remained as independent predictors of worse RFS (HR = 2.58, 95%CI 1.35–4.80, *p* = 0.0016, and HR = 3.30, 95%CI 1.40–6.24, *p* = 0.0060, respectively, Table 4).

## 4. Discussion

To the best of our knowledge, this is the first study to demonstrate the clinical implications of ΔSUVmax% in pancreatic cancer patients. We showed that the combination of SUVmax1 and/or ΔSUVmax% predicted tumor relapse within 1 year of surgery with higher sensitivity or specificity than SUVmax1 alone, especially in diabetes mellitus patients. We also found that the combination of SUVmax1 and ΔSUVmax% was an independent predictor of poor RFS.

In the present study, a history of diabetes mellitus was associated with a reduced SUVmax1 value, but not SUVmax% value. In addition, ΔSUVmax% had high sensitivity to predict early postoperative recurrence within 1 year compared to SUVmax1 in pancreatic cancer patients. Diederichs et al. [18] previously indicated that the presence of hyperglycemia (130 mg/dl) significantly reduced the SUV value of pancreatic cancer lesions. Our results suggest that the calculation of ΔSUVmax% reduces the influence of hyperglycemia and is more useful than the measurement of SUVmax1 alone to predict the outcomes of patients with diabetes mellitus.

In a recent randomized controlled trial that showed prolonged survival in patients receiving postoperative adjuvant chemotherapy with S-1 than those receiving gemcitabine (JASPAC01), the 5-year RFS rate of patients with S-1 was 33.3%. Among 129 events of recurrence or death, 62 (48%) occurred within 1 year of surgery [19]. Early postoperative recurrence, particularly within 1 year, generally indicates aggressive clinical features and has been associated with worse survival than recurrence after 1 year or longer [20]. Therefore, recurrence within 1 year of surgery was defined as early recurrence in the present study. The selection of pancreatic cancer patients at a high risk of early recurrence before surgery using DTP FDP PET/CT and the initiation of chemotherapy as the first-line treatment for at least several months followed by conversion surgery may improve the prognosis of these patients.

FDG-PET for pancreatic cancer is mainly used for tumor staging [21], the detection of recurrence after surgery [22], or monitoring the effects of chemotherapy and chemoradiotherapy [23,24,25]. Its utility as a prognostic predictor is not widely recognized. A few recent studies identified preoperative SUVmax1 as a significant predictor of early postoperative recurrence and subsequent poor survival following resection for pancreatic cancer [8,26]. We participated in the clinical trial by the Study Group of Preoperative Therapy for Pancreatic Cancer (PREP) to examine the usefulness of preoperative chemotherapy for resectable pancreatic cancer. In this study, we needed to perform PET/CT to exclude distant metastases before the patient enrollment [27]. This let us realize the utilities of this modality to find or confirm distant metastases that were difficult to recognize by CT or MRI. There is no randomized controlled trial in estimation costs for PET/CT, but PET/CT provided a significant incremental diagnostic benefit in the diagnosis of pancreatic cancer compared with CT alone and significantly influenced the staging and management of pancreatic cancer patients [28].

In the present study, 8 patients were excluded because of difficulties measuring SUVmax1 due to the low accumulation of ^18^F-FDG. Only 2 out of the 8 patients (25%) had recurrence. One of these patients was a 70-year-old male who did not receive adjuvant chemotherapy due to the comorbidity of chronic renal failure. Lung metastases appeared 10 months after surgery. The other patient was a 56-year-old female who developed lung metastasis 20 months after surgery, is alive, and has been receiving chemotherapy for 32 months. The other 6 patients have been alive without tumor recurrence for 22–74 months. Among the 8 patients excluded from the analyses due to low FDG accumulation, only 2 patients suffered from diabetes mellitus, and 5 patients (62.5%) had lymph node metastasis. Based on these results, we speculate that low accumulation of ^18^F-FDG might represent low malignant potential in resectable pancreatic cancer patients. The advantage of PET/CT is its ability to perform quantitative assessments [29]. The outcomes of 8 patients, the low accumulation of ^18^F-FDG, and our results in this study suggest the utility of FDG uptake to visualize tumor aggressiveness.

We typically evaluate SUVmax1 on PET/CT images, which is measured 60 min after the injection of ^18^F-FDG. We may obtain more detailed information on the precise biological nature of the target lesion from the later phase. Previous studies reported that ^18^F-FDG uptake in malignant lesions continued to increase until approximately 4–5 h after the injection, while it decreased in benign lesions 30 min after the injection [14,30]. It could be expected a higher number of cells with high uptake at the first point of detection, comprising both cancer cells and a specific signal from normal cells, while at the later time point the uptake would be assigned to only cancer cells. Thus, a decrease in false positives is assumed. Nowadays, diabetes mellitus patients with blood glucose levels of less than 200 mg/dl are still appropriate candidates to undergo PET/CT, as blood glucose levels of less than 200 mg/dl would not significantly change the tumor’s FDG uptake [31]. A correlation was previously reported between SUVmax% and malignant potential in lung cancer and lymphoma [15,16], but not in pancreatic cancer. We herein demonstrated that preoperative high SUVmax1 and ΔSUVmax% values were associated with an elevated risk of early postoperative recurrence and worse RFS in patients who underwent surgical resection for pancreatic cancer. We also showed that the combination of SUVmax1 and ΔSUVmax% more accurately predicted early recurrence than SUVmax1 alone.

The limitations of the present study include its retrospective nature and small sample number. Further studies with a larger number of patients are needed to validate the present results showing the usefulness of ΔSUVmax%. To demonstrate the efficacy of ΔSUVmax% in patients with diabetes mellitus, we needed to confirm that ΔSUVmax% was a better parameter than SUVmax1 in the subgroup of patients with diabetes mellitus; however, this was not possible because of the small number of patients. Nevertheless, the present study revealed the usefulness of ΔSUVmax% in patients, including those with diabetes. Among the volumetric parameter of PET/CT, metabolic tumor volume (MTV) and total lesion glycolysis (TLG) are popular. However, they are rarely mentioned in daily radiology reports because they cannot be measured as easily as SUVmax. Although DTP imaging with FDG PET/CT takes longer than other radiological imaging examinations, SUVmax parameters were easily assessed and reproducible. Inconvenience to patients is minimal.

## 5. Conclusions

DTP FDG PET/CT is a useful modality for predicting the early postoperative relapse of pancreatic cancer, even in diabetic patients. The combination of SUVmax1 and ΔSUVmax% more accurately identified a group of patients at high risk of early recurrence than SUVmax1 alone, and high SUVmax1 and high ΔSUVmax% were identified as independent prognostic factors.

## Figures and Tables

**Figure 1 cancers-14-03688-f001:**
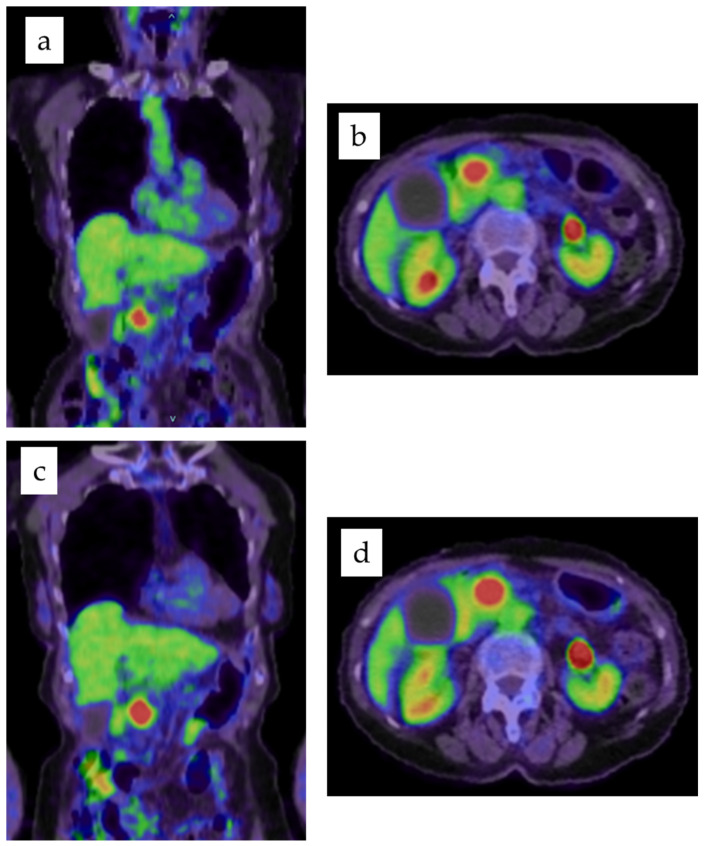
Maximum intensity (**a**) coronal and (**b**) axial images of the first scan in FDG PET/CT, and maximum intensity (**c**) coronal and (**d**) axial images of the second scan in FDG PET/CT. SUVmax1 was 8.9. SUVmax2 was 11.2.

**Figure 2 cancers-14-03688-f002:**
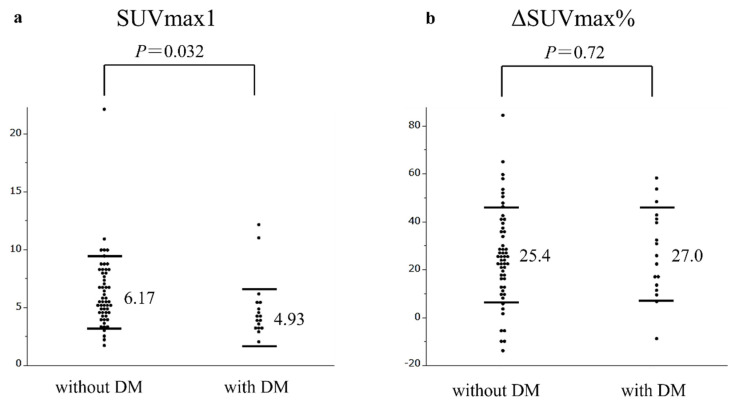
Comparison of SUVmax1 (**a**) and ΔSUVmax% (**b**) between patients with and without diabetes mellitus. (**a**) Patients with tumors showing without diabetes mellitus was significantly higher SUVmax1 than patients with tumor showing with diabetes mellitus (*P* = 0.032). (**b**) Patients with tumors showing without diabetes mellitus was not higher ΔSUVmax% than patients with diabetes mellitus (*P* = 0.72).

**Figure 3 cancers-14-03688-f003:**
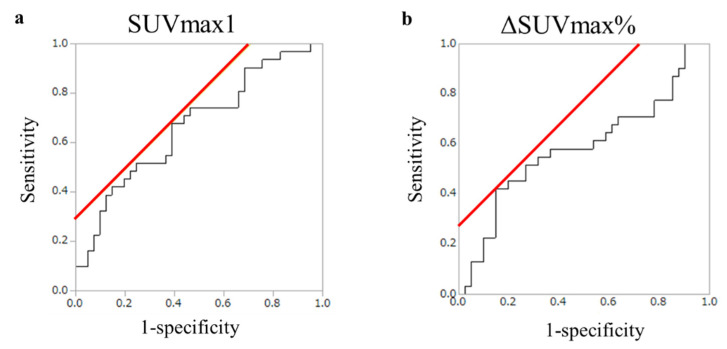
Selection of the cut-off point for and ΔSUVmax%. (**a**) Receiver operator characteristic (ROC) curves of the maximum standardized uptake value at 60 min (SUVmax1) with reference to relapse events within one year of pancreatectomy (*n* = 72). SUVmax1 at the cut-off value was 7.18, and the area under the curve (AUC) was 0.59 (95% confidence interval (CI) 0.44–0.72). (**b**) ROC curves of ΔSUVmax% with reference to relapse events within one year of pancreatectomy (*n* = 72). ΔSUVmax% at cut-off value was 24.25; AUC was 0.67 (95% CI 0.53–0.78).

**Figure 4 cancers-14-03688-f004:**
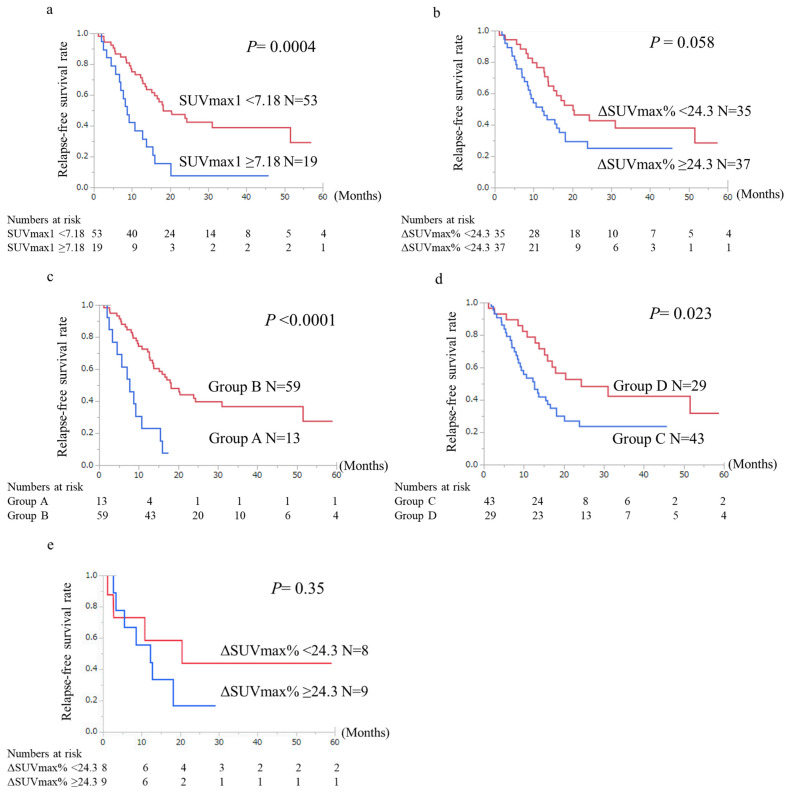
Comparison of relapse-free survival (RFS) curves between the patient groups of (**a**) high and low SUVmax1 values, (**b**) high and low ΔSUVmax%, (**c**) Group A, high SUVmax1 and high ΔSUVmax%, and Group B, and others, and (**d**) with Group C, high SUVmax1 and/or high ΔSUVmax%, and Group D, and others. (**e**) High and low ΔSUVmax% in diabetes mellitus patients. The group of high SUVmax1, Group A and Group C represented statistically unfavorable outcomes compared with the group of low SUVmax1 (**a**: *P* = 0.0004), Group B (**c**: *P* < 0.0001) and Group D (**d**: *P* = 0.023). On the other hand, ΔSUVmax% (**b**) and ΔSUVmax% in diabetes mellitus patients (**e**) were not correlated with RFS of the patients.

**Table 1 cancers-14-03688-t001:** Patient characteristics.

Parameter	Number of Cases
Age	
<70 years	30
>70 years	42
Median (range)	71 (86–50)
Sex	
Male	46
Female	26
Location	
Pancreatic head	50
Pancreatic body and/or tail	22
Pathological T-factor	
T1	2
T2	1
T3	68
T4	1
Pathological N factor	
Positive	58
Negative	14
Pathological M factor	
M0	69
M1	3
Residual tumor	
R0	62
R1	10
SUVmax	Median (range)
SUVmax1	5.1 (1.7–22.1)
SUVmax2	6.5 (1.9–25.6)
ΔSUVmax%	24.6 (−13.9–84.4)
Diagnosis of DM	
Yes	17
No	55
HbA1C Median (range)	5.9 (4.3–10.7)

SD, standard deviation; SUV, standardized uptake value; DM, diabetes mellitus.

**Table 2 cancers-14-03688-t002:** Accuracy of SUVmax1, ΔSUVmax%, and their combination for the prediction of relapse within 1 year of surgery.

Parameter	Number of Cases				Sensitivity	Specificity	PPV	NPV	Accuracy
	Total	1-Year Relapse	No Relapse	*p*	(%)	(%)	(%)	(%)	(%)
SUVmax1									
≥7.18	19	13 (68%)	6 (32%)	**0.0091**	41.9	85.4	68.4	66.0	66.7
<7.18	53	18 (34%)	35 (66%)						
ΔSUVmax%									
≥24.25	37	21 (57%)	16 (43%)	**0.0149**	66.7	61.0	56.8	71.4	63.9
<24.25	35	10 (29%)	25 (71%)						
Combination									
group A	13	10 (77%)	3 (23%)	**0.0060**	32.3	92.7	76.9	64.4	66.7
group B	59	21 (36%)	38 (64%)						
group C	43	24 (56%)	19 (44%)	**0.0068**	77.4	53.4	55.8	75.9	63.9
group D	29	7 (24%)	22 (76%)						

SUV, standardized uptake value; group A, high SUVmax1 and high ΔSUVmax%; group B, and others except A; group C, SUVmax1 and/or ΔSUVmax%; group D, and others except C. Values in bold are statistically significant.

**Table 3 cancers-14-03688-t003:** Relationships between SUVmax1, ΔSUVmax%, and clinicopathological parameters.

Parameter	No. of Cases N = 72	SUVmax1			ΔSUVmax%			Group A vs. Group B			Group C vs. Group D		
		High N = 19	Low N = 53	*p*-Value	High N = 37	Low N = 35	*p*-Value	Group A N = 13	Group B N = 59	*p*-Value	Group C N = 43	Group D N = 29	*p*-Value
Pathological T factor													
T1,2	3 (4%)	0 (0%)	3 (4%)	0.17	1 (1%)	2 (3%)	0.52	0 (0%)	3 (4%)	0.29	1 (1%)	2 (3%)	0.35
T3,4	69 (96%)	19 (26%)	50 (69%)		36 (50%)	33 (46%)		13 (18%)	56 (78%)		42 (58%)	27 (3 8%)	
Pathological N factor													
Positive	58 (81%)	8	40 (56%)	**0.044**	29 (40%)	29 (40%)	0.63	12 (17%)	46 (64%)	0.20	35 (47%)	23 (32%)	0.83
Negative	14 (19%)	1 (1%)	13 (18%)		8 (11%)	6 (8%)		1 (1%)	13 (18%)		8 (11%)	6 (8%)	
Lymphatic permeation													
Positive	64 (89%)	19 (26%)	45 (63%)	**0.022**	34 (47%)	30 (42%)	0.40	13 (18%)	51 (71%)	0.065	40 (56%)	24 (33%)	0.18
Negative	8 (11%)	0 (0%)	8 (11%)		3 (4%)	5 (7%)		0 (0%)	8 (11%)		3 (4%)	5 (7%)	
CA19-9													
>512	20 (28%)	9 (13%)	11 (15%)	**0.031**	12 (17%)	8 (11%)	0.36	8 (11%)	12 (17%)	**0.0043**	13 (18%)	7 (10%)	0.57
<512	52 (72%)	10 (14%)	42 (58%)		25 (35%)	27 (36%)		5 (7%)	47 (65%)		30 (42%)	22 (31%)	
Residual tumor													
R0	63 (88%)	16 (22%)	47 (65%)	0.62	32 (44%)	31 (43%)	0.79	10 (14%)	53 (74%)	0.23	38 (53%)	25 (38%)	0.79
R1	9 (13%)	3 (4%)	6 (8%)		5 (7%)	4 (6%)		3 (4%)	6 (8%)		5 (7%)	4 (6%)	
Completed adjuvant chemotherapy													
Yes	36 (50%)	8 (11%)	28 (39%)	0.42	12 (17%)	24 (33%)	**0.0019**	4 (6%)	32 (44%)	0.12	16 (22%)	20 (28%)	**0.0076**
No	36 (50%)	11 (15%)	25 (38%)		25 (38%)	11 (15%)		9 (13%)	27 (38%)		27 (38%)	9 (13%)	

SUV, standardized uptake value; PPV, positive predictive value; NPV, negative predictive value; group A, high SUVmax1 and high ΔSUVmax%; group B, and others except A; group C, SUVmax1 and/or ΔSUVmax%; group D, and others except C. Values in bold are statistically significant.

**Table 4 cancers-14-03688-t004:** Univariate and multivariate analyses of relapse in patients with pancreatic cancer.

Parameter (Favorable vs. Unfavorable)	Univariate		Multivariate			
	Hazard Ratio (95% CI)	*p*-Value	Including SUVmax1		Including SUVmax1/ΔSUVmax%	
			Hazard Ratio (95% CI)	*p*-Value	Hazard Ratio (95% CI)	*p*-Value
Pathological T-factor	1.59	0.62				
(pT3,4 vs. T1,2)	(0.35–28.22)					
Pathological N-factor	3.91	**0.0018**	2.51	0.062	2.90	**0.026**
(Positive vs. Negative)	(1.58–13.03)		(0.96–8.59)		(1.12–9.88)	
CA19-9	2.67	**0.0023**	1.78	0.074	1.48	0.24
(>512 vs. <512)	(1.44–4.83)		(0.94–3.28)		(0.76–2.80)	
Residual tumor	1.83	0.15				
(R1 vs. R0)	(0.79–3.73)					
Completed adjuvant chemotherapy	2.94	**0.0003**	2.59	**0.0016**	2.48	**0.0027**
(No vs. Yes)	(1.64–5.47)		(1.43–4.85)		(1.36–4.63)	
SUVmax1	2.95	**0.0011**	2.58	**0.0016**		
(≥7.2 vs. <7.2)	(1.57–5.39)		(1.35–4.80)			
SUVmax1/ΔSUVmax%	3.82	**0.0006**			3.03	**0.0060**
(group A vs. group B)	(1.84–7.49)				(1.40–6.24)	

CI, confidence interval; SUV, standardized uptake value; group A, high SUVmax1 and high ΔSUVmax%; group B, and others except A. Values in bold are statistically significant.

## Data Availability

All data generated and analyzed during this study can be retrieved by sending a formal request by email to the corresponding author.

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
