# Peer review of "Clinical Impact of Dual Time Point 18F-Fluorodeoxyglucose Positron Emission Tomography/Computed Tomography Fusion Imaging in Pancreatic Cancer"

_cancers, 2022, doi:10.3390/cancers14153688_

Round 1
Reviewer 1 Report
The work by Einama et al showed interesting results on the possible introduction of DTP FDG PET/CT as a modality for predicting early recurrence in patients subjected to surgical resection for pancreatic cancer. In particular, authors proposed that the technique is more informative than the single FDG PET/CT, allowing to retrieve also the information on the ΔSUVmax%. This value account a greater number of patients that will encounter relapse. Also, it could overcome the measurements problem in patients with diabetes mellitus.
They observed that the combination of SUVmax1 and ΔSUVmax% show higher specificity than SUVmax1 alone and increase the prediction of patients with worse relapse-free survival rate.
The overall study is of potential usefulness. However, the work results superficial, almost identical to a previous published paper from some of the authors and the lack of clarity in the presentation of the results impact on the quality of the presented paper.
MAJOR POINTS
The introduction about FDG PET/CT and DTP FDG PET/CT is poorly written. It is very generic and chaotically organized. First, the techniques should be deeper explained. The molecular mechanism at the basis of this type of measurement and its detection has to be described better. Second, the disadvantages of the single measurement has to be pointed out and, then, the improvement of the dual methodology in respect to the single one should be clearly elucidated. Please adjust and expand this part of the introduction.
Tables 2 and 3 are probably been inverted by the authors. Otherwise, please indicate clearly in the Table 2 and 3 the data described in the results.
Please explain or speculate why you see an increased number of patients with high ΔSUVmax% in respect to those with high SUVmax1. It could be expected a higher number of cells with high uptake at the first point of detection, comprising both cancer cells and aspecific signal from normal cells, while at the later time point the uptake would be assigned to only cancer cells. Thus, a decrease of false positives is assumed.
Although the small number of patients with diabetes mellitus in the study, would be useful for the significance of the proposed article to investigate how the ΔSUVmax% could implement the prediction of recurrence in those patients. The authors should add this data to the results.
The authors reported that “PET/CT is costly and only available at few institutions”. The reviewer was wondering how the DTP FDG PET/CT could be effectively proposed as a routinary exam in clinics for pancreatic cancer patients. Please discuss this aspect.
In general, the conclusions retrieved from each results paragraph are not reported and also the title of the paragraph is not representative. Please, provide more descriptive interpretation.
MINOR POINTS
The introduction on pancreatic cancer has to be expanded and more punctual references provided.
Patient selection: specify for how long the postoperative surveillance was performed.
Provide a reference for the specified dosage used of 18F-FDG.
Explain why the two scans in PET/CT focused on different regions.
Describe the parameters used for PET/CT e.g., slice thickness.
Specify how the region of interest is defined.
If a specific software was used to analyze the images obtained from PET/CT, cite which one.
Report T, N and M factors definition.
Explain more clearly the strategy of groups division.
Specify which statistical analysis is used for every comparison throughout the text.
About the patients excluded from the analyses, speculate on the reason why low accumulation of 18F-FDG occurred.
Please change the yellow color in the ROC curves in Fig2 to render the results more visible and clearer.
In the comparison of clinicopathological parameters deeper describe the four different approaches used.
Introduce the comparison of survival paragraph in a proper manner and highlight the conclusions authors retrieved.
In the discussion authors report “The outcomes of these patients also suggest that FDG uptake, SUVmax1, is strongly associated with tumor aggressiveness in pancreatic cancer.” This speculation has to be clearly explained.
Lines 15-17: check grammar.
Lines 20-22: check grammar. Repetition.
Lines 22-23: check grammar.
Lines 63-67: the aim of the study is quite redundant.
Lines 134-135: check grammar.
Line 155: check grammar.
Lines 208-210: check grammar. The authors reported:” … and ΔSUVmax% predicted early postoperative recurrence within 1 year more accurately than SUVmax1.” Seems that is referred to diabetes mellitus patients, but this is not supported by the present study.
Author Response
Thank you for your advise. I have revised my manuscript according to your suggestions.

Reviewer 2 Report
This article elucidated the roles of of preoperative dual time point (DTP) 18F-fluorodeoxyglucose positron emission tomography/computed tomography fusion imaging (FDG PET/CT) as a predictor of early recurrence in or the outcomes of patients with pancreatic cancer. The result is of interests and shows that the combination of SUVmax1 and ΔSUVmax% identified early recurrent patient groups more precisely than SUVmax1 alone, and it can overcome the drawback of FDG PET/CT of the false low tumor SUV in patients with hyperglycemia. However, the method of DTP FDG PET/CT is more conventional, without innovation, and has the disadvantage of excessive radiation.The authors did well in designing and writing the manuscript. To improve this manuscript, some small points may be noted as follows:
1.Line 55-57: “Another drawback of FDG PET/CT is the false positive accumulation of SUV in patients with hyperglycemia. Hyperglycemia is associated with significantly reduced 18F-FDG up-take levels”. I don't think this accurately expresses the meaning of the quotation. It should be that another drawback of FDG PET/CT is the false negative accumulation of SUV in patients with hyperglycemia. Hyperglycemia is associated with significantly reduced 18F-FDG up-take levels or another drawback of FDG PET/CT is the false low SUV in patients with hyperglycemia. Hyperglycemia is associated with significantly reduced 18F-FDG up-take levels. This view of Low tumoral FDG uptake of patients with markedly elevated plasma glucose levels comes from a study in 1998, so it is recommended to quote the updated research results.
2.Line 157-178: The positions of Table 2 and table 3 in the description of research results are reversed
3.Line 159-3161: The analysis of the statistical results of the four groups in Table 3 is not detailed enough. The results should show that group A and group C have statistical significance in identifying recurrence, and then explain the sensitivity, specificity, PPV, NPV difference between group A and group B.
Author Response
Thank you for your comments. I have revised my manuscript according to your suggestions.

Reviewer 3 Report
Imaging modalities are being extensively studied to potentially help the prediction of therapy outcomes and the guidance of clinical decisions. This work focused on PET/CT scan, particularly 18F-FDG PET/CT, which is commonly used to evaluate sites with active glucose metabolism and reported that a combination of SUVmax1 and deltaSUVmax% identified early recurrent patients more precisely than the SUVmax1 alone.
Several questions are listed below regarding this manuscript.
1. In abstract, how long the relapse-free survival (RFS) is able to be predicted most accurately using the method of this work? Is it 1 year?
2. Survival curves need “number at risk”.
3. Are there representative imaging data that can be presented in the main text?
4. Why 18F-FDG were selected for this study?
5. How are the SUVmax values considered to be used for machine learning model building for future application?
6. Lack of recent literature.
Author Response

(The authors gave the same response as above.)

Round 2
Reviewer 1 Report
The manuscript by Einama et al. showed significant improvements.
Among the requests asked by the reviewer, is still not addressed:
- The authors reported that “PET/CT is costly and only available at few institutions”. The reviewer was wondering how the DTP FDG PET/CT could be effectively proposed as a routinary exam in clinics for pancreatic cancer patients. Please discuss this aspect.
The sentence introduced by the authors does not discuss the fact that is difficult to imagine that DTP FDG PET/CT could be introduced as a routinary exam due to its expensiveness, as reported by the authors.
- About the patients excluded from the analyses, speculate on the reason why low accumulation of 18F-FDG occurred.
The authors completely missed to answer this point.
- In the comparison of clinicopathological parameters deeper describe the four different approaches used.
The authors completely missed to answer this point.
- In the discussion authors report “The outcomes of these patients also suggest that FDG uptake, SUVmax1, is strongly associated with tumor aggressiveness in pancreatic cancer.” This speculation has to be clearly explained.
When the authors is referring to "these patients" I was wondering if they are speaking about the 8 patients excluded from the study. In this case, the corrected sentence is still not clear to me.
- Lines 22-23: check grammar, the sentence is missing the subject and the verb.
- Lines 134-135: check grammar, "with than".
- Lines 208-210: check grammar, "... and ΔSUVmax% predicted early postoperative recurrence within 1 year more accurately than SUVmax1." This is a general conclusion or is referring only to diabetes mellitus patients?
Author Response
Thank you for your advice.
I have revised my manuscript according to your suggestions.

Reviewer 3 Report
The manuscript has improved a lot.
A remaining question is :
In figure 4, how come data points (6) of number at risk does not align with the plot x-axis (7)? and how come in fig4c, number at risk in group A increased?
Author Response

(The authors gave the same response as above.)
